# The Surgical Risk Factors of Giant Intracranial Meningiomas: A Multi-Centric Retrospective Analysis of Large Case Serie

**DOI:** 10.3390/brainsci12070817

**Published:** 2022-06-22

**Authors:** Daniele Armocida, Antonia Catapano, Mauro Palmieri, Umberto Aldo Arcidiacono, Alessandro Pesce, Fabio Cofano, Veronica Picotti, Maurizio Salvati, Diego Garbossa, Giancarlo D’Andrea, Antonio Santoro, Alessandro Frati

**Affiliations:** 1Neurosurgery Division, Human Neurosciences Department, “Sapienza” University, 00135 Rome, RM, Italy; antoniacatapano2@gmail.com (A.C.); mauro.palmieri@uniroma1.it (M.P.); arcidiaconomd@gmail.com (U.A.A.); antonio.santoro@uniroma1.it (A.S.); 2IRCCS “Neuromed”, Neurosurgeon Consultant, Via Atinense, 18, 86077 Pozzilli, IS, Italy; alex.frati@gmail.com; 3Neurosurgery Unit, Ospedale Santa Maria Goretti, Via Guido Reni 1, 04100 Latina, LT, Italy; ale_pesce83@yahoo.it; 4Neurosurgery Unit, Department of Neuroscience “Rita Levi Montalcini”, University of Turin, Via Cherasco, 15, 10126 Torino, TO, Italy; fabio.cofano@gmail.com (F.C.); diego.garbossa@unito.it (D.G.); 5Neurosurgery Department of Fabrizio Spaziani Hospital, Via Armando Fabi, 03100 Frosinone, FR, Italy; veronica@picotti.com (V.P.); gdandrea2002@yahoo.it (G.D.); 6Policlinico Tor Vergata, University Tor Vergata of Rome, Viale Oxford, 81, 00133 Roma, RM, Italy; salvati.maurizio@libero.it

**Keywords:** meningioma, brain tumor, peritumoral brain edema, giant meningiomas

## Abstract

Giant intracranial meningiomas (GIMs) are a subgroup of meningiomas with huge dimensions with a maximum diameter of more than 5 cm. The mechanisms by which a meningioma can grow to be defined as a “giant” are unknown, and the biological, radiological profile and the different outcomes are poorly investigated. We performed a multi-centric retrospective study of a series of surgically treated patients suffering from intracranial meningioma. All the patients were assigned on the grounds of the preoperative imaging to giant and medium/large meningioma groups with a cut-off of 5 cm. We investigated whether the presence of large diameter and peritumoral brain edema (PBE) on radiological diagnosis indicates different mortality rates, grading, characteristics, and outcomes in a multi-variate analysis. We found a higher risk of developing complications for GIMs (29.9% versus 14.8%; *p* < 0.01). The direct proportional relationship between PBE volume and tumor volume was present only in the medium/large group (Pearson correlation with *p* < 0.01) and not in the GIM group (*p* = 0.47). In conclusion, GIMs have a higher risk of developing complications in the postoperative phase than medium/large meningioma without higher risk of mortality and recurrence.

## 1. Introduction

Meningiomas represent one-third of all are primary central nervous system (CNS) tumors in adults with a female prevalence and median age at diagnosis of 66 years old [1]. They are typically benign and arise from meningothelial cells. Most meningiomas are slow-growing lesions with a growth rate of approximately 2.4 mm per year [2]. For many patients who present with meningioma, in particular asymptomatic tumors, observation with routine surveillance imaging alone is an acceptable strategy while for tumors that are growing or causing symptomatology, maximal safe surgical resection remains the standard of care for therapeutic management. The clinical manifestations depend on their location and grade of their mass effect, but some tumors may grow over time without giving any clinical symptoms [3] and therefore debut with a considerable size [4,5,6,7]. Giant intracranial meningiomas (GIMs), defined as contrast-enhancing lesions with a maximum diameter of more than 5 cm, are uncommon and are usually considered arduous to resect totally with a poorer prognosis [8,9,10,11,12,13]. Further, GIMs are associated with different degrees of peritumoral brain edema (PBE) that represents one of the major causes of poorer prognosis [14,15,16,17]. It is not unusual to observe GIMs with a large variety of extensions of PBE. Several studies on GIMs were reported previously, but many are case reports or small case series [3,11,18,19,20,21]. The real mechanisms by which a meningioma can grow to be defined as “giant” are unknown, as well as the real biological and radiological profile and the different outcomes that a patient treated surgically for such an infrequent form may have. We present a series of 340 cases who underwent surgical management of primary intracranial meningioma, analyzing clinical, radiological, and pathological characteristics, and we evaluated outcome and risk rate on the grounds of size (117 GIMs and 223 medium/large meningiomas). We focused on the surgical challenges of this uncommon presentation of tumor and highlighted the radiological, histological, and anatomical characteristics, and surgical techniques intending to remove the most important risk factors of the outcome.

## 2. Methods

### 2.1. Participants and Eligibility

We performed an institutional retrospective review of a consecutive series of surgically treated patients suffering from histologically confirmed intracranial meningioma, operated on in the Sapienza Neurosurgery Department of Rome (Italy) and Neurosurgery Department of Hospital Spaziani of Frosinone (Italy) in the period ranging between January 2016 and December 2020.

We collected a total of 472 patients suffering from meningioma. We adopted the following inclusion and exclusion criteria: -Patients with confirmed histological diagnosis of meningioma performed according to the updated version of the 2021 WHO guidelines [22] at their first surgery;-patients were included in the study if their pre- and postoperative magnetic resonance imaging (MRI) was either performed at our institution or available on the picture archiving and communication system (PACS) for review;-patients were included if, in the postoperative period, could undergo a standard clinical and radiological follow-up starting from the 30th day after surgery; patients were excluded for incomplete or wrong data in clinical, radiological, and surgical records and/or being lost to follow-up;-the estimated target of the surgical procedure was the total or subtotal resection of the lesions; no biopsies were included.

All the patients who met the aforementioned inclusion criteria were assigned on the grounds of the preoperative imaging to the following subgroups:
-Tumors classified as giant meningiomas (Group A): The contrast-enhanced lesion measured at least 5 cm along the major diameter in T1-weighted images with MRI;-tumors classified as medium/large meningiomas (Group B): The contrast-enhanced lesion measured less than 5 cm along the major diameter in T1-weighted images with MRI.

In defining the size of meningiomas, we referred to the traditional classification of Russell et al. [23] which reports small meningiomas as those less than 2 cm in diameter, medium as 2 to 4 cm, and large up to 4.9 cm.

Medium/large meningiomas were considered as one group for this study because they have similar prognostic features and are often variably defined in many studies [1,3,14,15,24,25,26,27,28,29,30,31].

Being a surgical collection, we included only cases with an indication for treatment and who underwent surgery. Patients with small meningiomas were sent for follow-up control or radiosurgery. Patients with meningiomas of 2 cm was submitted to a surgery if symptomatic or if they demonstrated failure in radio surgery or medical therapy for seizure.

For all the included patients, we recorded at first: Age, sex, time of hospitalization, time of follow-up, clinical onset, presence of smoking habits, hypertension and performance status (measured using Karnofsky performance scale (KPS)) at the moment of radiological diagnosis.

We categorized all included patients with comorbidity using the Charlson comorbitiy index (CCI).

Regarding the clinical onset, we considered as focal neurological deficits the focal disorders of body motility and sensitivity, sphincter disorders, and disorders involving cranial nerves including visual disturbances. We also considered the presence of dizziness, alteration of mental status and memory loss, the presence of intractable headache, seizure, and the incidental diagnosis.

All included patients were evaluated by a neuroradiologist to consider the possibility of adjuvant embolization of feeders [32,33].

### 2.2. Imaging Analysis and ROI-Drawing Process

All the patients included underwent a preoperative brain MRI scan including a high field 3 Tesla volumetric study. On radiological evaluation, we recorded location of the lesion, the presence of multiple meningiomas and or meningiomatosis, the involvement of subtentorial compartment, tumor major diameter (measured in cm), and tumor volumes (measured in cm^3^) using isotropic volumetric T1-weighted sequences before and after intravenous administration of paramagnetic contrast agent (gadolinium). We used T2-weighted and fluid attenuated inversion recovery (FLAIR) sequences to obtain the edema volumes (measured in cm^3^ before anti-edemigen therapy). 

Volume of the contrast-enhanced lesion and edema were calculated by drawing a region of interest (ROI) in a volumetric enhancing postcontrast study weighted in T1 (a multi-voxel study) and T2 using the software Horos.

In 3D T1-weighted contrast-enhanced sequences, we used a pencil-draw semi-automatic tool to outline the lesion on all subsequent slices of a minimum of 1.00 mm thickness conforming to the margins of the contrast-enhanced lesion with the software Horos.

With the same method, using T2-weighted images and FLAIR, we outlined the hyperintensity area defined as “the high signal on T2-weighted imaging of brain sequences” referring to the entire volume including the tumor lesion and associated tumor mass to obtain a new ROI (Figure 1).

Once obtaining a volumetric ROI of the tumor mass (Figure 2), with a simple arithmetic subtraction between these two volumes we obtained the value in cm^3^ of the perilesional edema of the tumor [34,35].

### 2.3. Surgical Treatment

The patients underwent several transcranial and skull base approaches according to the site of the meningioma. Olfactory groove (OG) and anterior floor lesions were removed through supraorbital, cranio-orbital, and supra-orbital bi-frontal approaches; spheno-orbital and temporal floor meningiomas were removed via the cranio-orbital zygomatic approach; sphenopetroclival (SPC) meningiomas were removed through the anterior or posterior petrosal approach; and tentorial meningiomas were removed through the suboccipital and/or retrosigmoid approaches.

On the first postoperative day, the patients underwent a volumetric brain MRI scan to evaluate the extent of resection (EOR) and measure the Simpson grade. Every patient with Simpson grade over I and WHO type II and III was submitted to radiotherapic and oncological evaluation.

Radiotherapy treatment has been reserved for surgically treated patients with grade III, atypical, and grade II meningioma with residual tumor at first MRI examination and meningiomas with a large component considered “unresectable” for location and operative risks. The treatment scheme performed by our center was according to the NCCN guidelines version 3.2020 of 59.4 Gy total in multi-fraction regimen of 1.8 Gy in 33 fractions for grade II meningiomas, and 60 Gy in multi-fraction regimen of 2 Gy in 30 fractions for grade III meningiomas [36].

On the grounds of the histological final diagnoses, we recorded: WHO grading with subtypes, mitotic index measured using the count of mitosis on 10 HPF, immunohistochemistry with Ki-67 and progesteron (PR) expression routinely performed in the Department of Neuropathology of our hospital; Ki-67 was applied to frozen sections of fresh tissue using a standard immunoperoxidase technique.

Overall survival (OS) was recorded in months; it was measured from date of diagnosis to date of death or date of last contact if alive. Clinical information was obtained from the digital database of our institution, whereas OS data were obtained by telephone interview. We recorded after the surgical procedure the status of performance (using KPS) for each patient at 1 month, 6 months, and at last clinical evaluation. There was a special focus on the KPS results: This parameter was considered, as previously observed, as predictive and associated with survival (methodology described for other studies reported [37,38]). We evaluated the presence of complications, recurrence, and consequent second treatment recording biological switch. We investigated whether the presence of a large diameter on radiological diagnosis is indicative for different OS, grading, immunohistochemical characteristics, and clinical/neurological outcome.

### 2.4. Statistical Methods

The sample was analyzed with SPSS version 18. Comparisons between nominal variables were made with a chi^2^ test. EOR (measured with Simpson grade) means were compared with one-way and multi-variate ANOVA analysis along with contrast analysis and post hoc tests. Continuous variable correlations were investigated with Pearson’s bi-variate correlation. Threshold of statistical significance was considered *p* < 0.05.

### 2.5. The Potential Source of Bias and Study Size

We addressed no missing data since incomplete records were excluded. A potential source of bias is expected to derive from the exiguity of the sample, which nevertheless, in regard to the endpoints selected, presents an excellent post hoc statistical estimated power (difference between two independent means; 1- β = 0.9488 for α 0.05 and effect size 0.5), thus providing extremely reliable conclusions.

The informed consent was approved by the Institutional Review Board of our institution (IRB 6168 Prot. 0935/2020). Before the surgical procedure, all the patients gave informed written explicit consent after appropriate information. Data reported in the study have been completely anonymized. For statistical analysis, data collection, and analysis of results we have received support from the Neurosurgical Department of Turin, Italy directed by Prof. D. Garbossa. No treatment randomization has been performed due to the study’s retrospective nature. This study is consistent with the Helsinki Declaration of ethical principles for medical research on humans.

## 3. Results

### 3.1. Descriptive Data

The final cohort consisted of 340 patients (102 males and 238 females—70% of the population) respecting the F: M ratio reported in the literature of 2–3:1 [1,2]. The average age was 60.38 ± 13.56 years (min: 20, max: 90); Smoking habits and hypertension were revealed at the time of radiological diagnosis in 98 (28.8%) and 108 patients (31.8%), respectively. Patient selection is reported in Figure 3.

We report the clinical debut for all included cases (Table 1); although only symptomatic meningiomas or meningiomas large enough to be evaluated as surgical were considered in this collection, and a significant percentage of patients (14 patients, 13.2%) were incidentally diagnosed after investigations for other pathologies.

In a final division in the main Group A, there were 117 giant meningiomas (34.4%) and, in Group B, there were 223 medium/large meningiomas. The two subgroups did not present remarkable differences from the age/sex differences. Clinical debut, presence of seizure (*p* = 0.76), smoking habits (chi-square = 1.362; Df = 1; *p* = 0.24), and hypertension (chi-square = 1.4; dF = 1; *p* = 0.24) were not significantly associated. The comorbidity status measured as CCI did not reveal significant differences between groups. All the relevant details with analysis results are included in Table 2.

### 3.2. Histochemical Comparison Analysis between the Two Groups

From the histochemical point of view, the two subgroups regarding the WHO classification presented significant differences. A significant relationship (chi-square = 24.05; dF = 1; *p* < 0.01) is shown between WHO grade, type (particularly atypical meningiomas), and tumor size.

Group A presented with a higher significant percentage of grade II (31 patients, 26.5% versus 16 patients, 7.2%; *p* < 0.01). 

Notably, there is a more evident difference between WHO grade I and II (*p* < 0.01) than between grade II and grade III (*p* = 0.82) in the growth rate. These data are confirmed with an independent association between Ki-67 expression and total tumor volume (p = 0.02).

There is no correlation between the expression of progesteron and the size of meningiomas in both groups, and this finding is confirmed when comparing the total volume of the lesion (*p* = 0.85) and the largest diameter of the tumor (*p* = 0.66).

### 3.3. Radiological Comparison Analysis between the Two Groups

The extent of cerebral edema in relation to tumor size was evaluated. As expected, Group A presented at radiological diagnosis with a higher edema volume (mean: 42.52 cm^3^ SD: 52.77) than Group B (mean: 18.37 cm^3^, SD: 38.59. *p* < 0.01).

We found that a directly proportional relationship between edema volume and tumor volume was present only in Group B (Pearson correlation, *p* < 0.01, Figure 4) with a statistically significant difference in proportionality, and this was not present in Group A (Pearson correlation, *p* = 0.74, Figure 5).

This finding is confirmed comparing the edema volume/lesion volume ratio (Pearson correlation, t = 2.44, dF = 214, *p* = 0.016). Results are obtained after Tukey and Bonferroni correction methods.

This peculiar feature suggests that GIMs compared with medium and large meningiomas have a less remarkably strong relationship between the volume of the tumor mass and the edema generated around the tumor in the brain tissue. We performed a multi-variate analysis comparing these data with grading and location; just a weak association (*p* = 0.17) between edema volume and grading is shown for both groups (Figure 6) and, analyzing the edema volumes among the different localizations of all meningiomas of this series, there is a variability among the different groups of meningiomas (ANOVA test, *p* = 0.04), with a prevalence of edematous lesions in meningiomas of the olfactory shower and sphenoidal plenum, without, however, substantial significance (p = 0.659, group 6, *p* = 1, group 11). In conclusion, there are no significantly edematous locations compared to others, but in medium/large meningiomas, as well as in the whole collection, there is no site more edematous than another, while in GIMs there is a mild correlation with the site of implantation as far as tumors of the anterior basicranium (olfactory shower and sphenoidal plenum) are concerned (this aspect will be evaluated in a further and dedicated study [35]).

### 3.4. Outcome Data and Main Results

Neurological and clinical outcomes were measured with KPS score for the entire collection and for the two subgroups. 

In general, the surgical outcome of meningiomas is affected by lesion localization, but to a different extent than recovery time. When comparing the performance status before and after surgery, there is a statistically significant difference between localization and KPS (*p* = 0.04), particularly evident for sphenopetroclival meningiomas (*p* = 0.07), and partially with meningiomas of the OG with arterial encasement (Figure 7 and Figure 8).

This difference is no longer evident in the comparison at the last evaluation where the final KPS has no correlation with the location of the meningioma (*p* = 0.32).

If we consider in the multi-variate analysis how performance status changes during follow-up, we observe that Group A presents a slower recovery and a stably lower KPS than Group B in a manner independent of clinical onset, age, and tumor location (*p* = 0.04, Figure 9).

It is found that surgically treated GIMs have a higher risk of developing complications in the postoperative phase (chi-square = 11.121; dF = 1; *p* < 0.01, Figure 10). The most frequently encountered complications include the occurrence of ischemia (*p* = 0.049), infection (*p* = 0.03), and especially the occurrence of postoperative seizures.

Although there is no evidence of a greater presence of epilepsy at diagnosis of a GIM compared to a medium/large meningioma (chi-square = 0.090; dF = 1; *p* = 0.764), there is an increased risk of seizures in the postoperative phase (chi-square = 8.555; dF = 1; *p* < 0.01).

On the other hand, there is no significant relationship (chi-square = 2.189; dF = 1; *p* = 0.14) between mortality and the presence at diagnosis of a GIM. In our case series, the risk of recurrence measured at the last evaluation was superimposable between Group A and Group B (chi-square = 2.581; dF = 1; *p* = 0.108).

## 4. Discussion

Although there is no exact definition of GIM in the literature (some authors defined GIM as a tumor of >4.5–5-6 or 7 cm in maximum diameter [28,29,30]), we accepted 5 cm as the lower limit of diameter reported in almost all large series [1,3,24,25,26,27,28,29,30,31]. 

The reason why a meningioma can grow out of proportion without giving symptoms regardless of intracranial location is not known [29,30,31,37,38,39]. 

We confirm that WHO grade II and atypical meningiomas more frequently give this type of radiological finding than malignant grades [40], which because of their infiltrative pattern may give symptoms earlier and thus may be diagnosed earlier [41,42]. We confirm that grading (especially atypical forms) [43] influenced by a high proliferation index [12] with high Ki-67 and number of mitoses per field is definitely more suggestive of a high growth rate than the presence of hormone-dependent forms that better explain the higher incidence in the female population and cases of meningiomatosis.

From the therapeutical perspective, while treatment for small/medium meningiomas is highly individualized and includes a combination of observation, surgical resection, and/or radiotherapy, for GIMs surgical treatment is considered the primary therapy for their mass effect and neurovascular involvement although it is associated with a confirmed higher risk of complications and mortality [19,43]. The study by Narayan et al. [28], which was until now the largest series with 80 cases of GIMs, demonstrated that regression analysis showed age, sex, location of the tumor [44,45], Simpson grade of excision, and histology of tumor were the factors that significantly affected the KPS, complications, and recurrence [39,46]. Our series confirms just some of these findings whereas the mortality rate, recurrence rate, Simpson grade, and KPS are comparable between the two groups, but there is a higher rate of complications in the first 30 days after the surgical procedure for GIMs compared with smaller meningiomas [47,48,49,50]. We identified a higher incidence of postoperative ischemia, infections, and seizure in the GIM group. These findings differ from other studies [18,30,46] where complications such as hemorrhage and malignant postoperative edema were more frequently identified with a stronger correlation with mortality.

One of the major factors that lead meningiomas to have more complications and reduced performance status is the volume of PBE and it is well-noted that a variable amount of vasogenic edema is shown in adjacent brain tissue in more than half of meningioma cases [31]. Edema could exacerbate neurological symptoms and increase morbidity and risk of postoperative complications [14,15]. The cause of the high frequency of cerebral edema in meningiomas has been much discussed and a variety of causative factors have been investigated [49], however, the exact mechanism of development of PBE remains unclear [14,15,16,17].

In common clinical practice, finding meningiomas of huge sizes with insignificant amounts of edema is usual. In our collection, we demonstrate that, although GIMs have more edema volume than medium-sized meningiomas, the correlation between tumor volume and edema is valid only up to a certain size and that for GIMs this relationship is no longer evident. Although the most intuitive hypothesis would be that a larger meningioma with a higher growth velocity and more frequently higher grade would result in a higher proportion of edema with related symptomatology (therefore, tumors associated with extensive PBE are commonly suspected to be pathologically malignant [17]), this does not occur. Further, although there is a correlation between grading with survival and recurrence rates [3,24,25], we did not find a correlation between PBE and grading.

The reasons for worse outcome of GIMs must be further investigated in other factors such as location and vascular encasement.

In our series, we have found a strong correlation with location (in particular with SPC GIMs) and in terms of predicting surgical complications, confirming previous data [30], and confirming that a cerebral artery encasement is associated with greater risk profiles (confirmed with the high risk of OG GIM, Figure 11) [37].

In any case, the surgery of GIMs is considered unique due to prominent vascularity, entangling, and limited visualization of various neurovascular structures and severe cerebral edema [46]. 

## 5. Limitations and Further Studies

The main limitation of this study is its retrospective nature, which does not allow for an effective risk study by randomization. In addition, an ad hoc molecular prognostic study should be performed on these types of tumors. Theories on the development of meningiomas and PBE also include multiple molecular factors such production of vascular endothelial growth factor (VEGF), and interleukin-6 expression, but further research needs to be done to understand the clinical behavior of these tumors. 

## 6. Conclusions

The study confirms that surgically treated GIMs have a higher risk of developing complications in the postoperative phase, but this risk appears to be determined by location (petroclival region and anterior basicranium) and the greater presence of arterial encasement rather than the presence of PBE. GIMs present with a greater volume of PBE than medium/large meningiomas, but with a poor proportional relationship with the amount of edema. In the tumors under 5 cm in maximum diameter, there is a stable relationship between the increase in the mass volume and the simultaneous increase in the volume of edema but this relationship is not found in GIMs. Although preoperative edema volume may represent a significant marker of poor functional outcome, this factor seems not relevant in GIMs.

## Figures and Tables

**Figure 1 brainsci-12-00817-f001:**
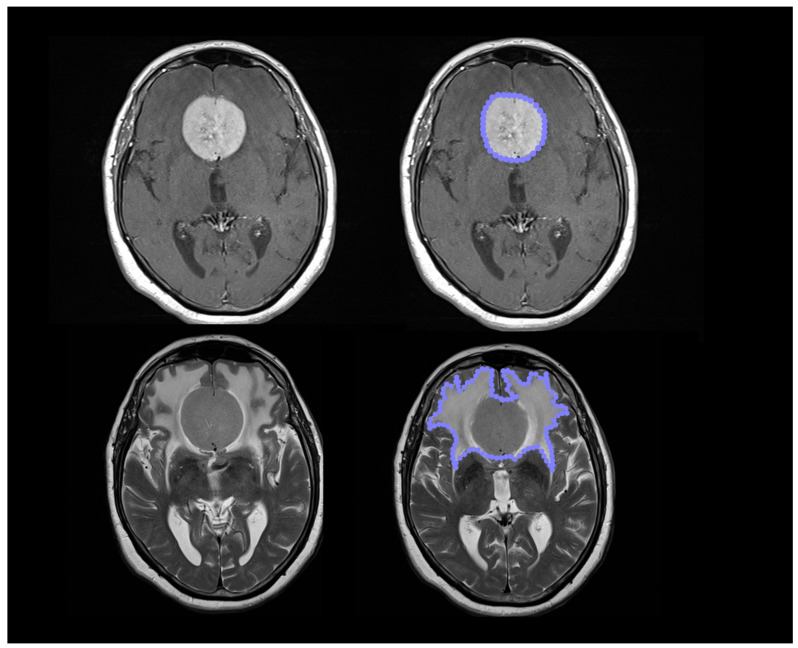
Using Horos software, we measured the volume of the meningioma using a semi-automatic pencil tool outlining the contrast-enhanced lesion on 3D T1-weighted sequences and its relative hyperintense signal T2-weighted sequences.

**Figure 2 brainsci-12-00817-f002:**
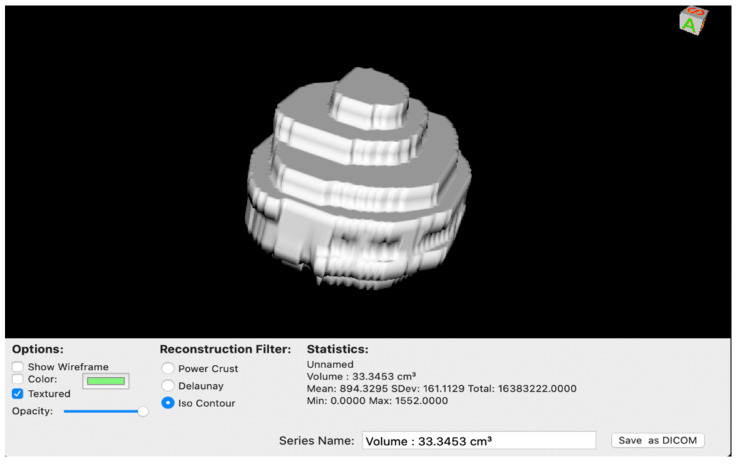
The ROI volume of the tumor was rendered in a 3D model before subtracting from T2-weighted sequences to obtain the volume of the edema.

**Figure 3 brainsci-12-00817-f003:**
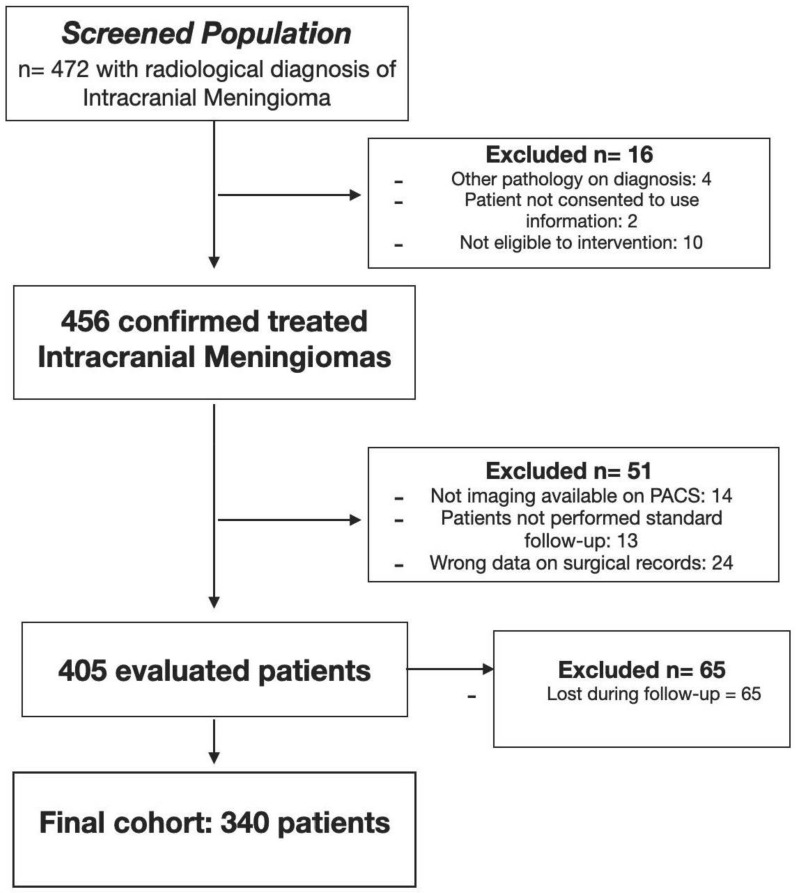
Flow-chart of patient selection.

**Figure 4 brainsci-12-00817-f004:**
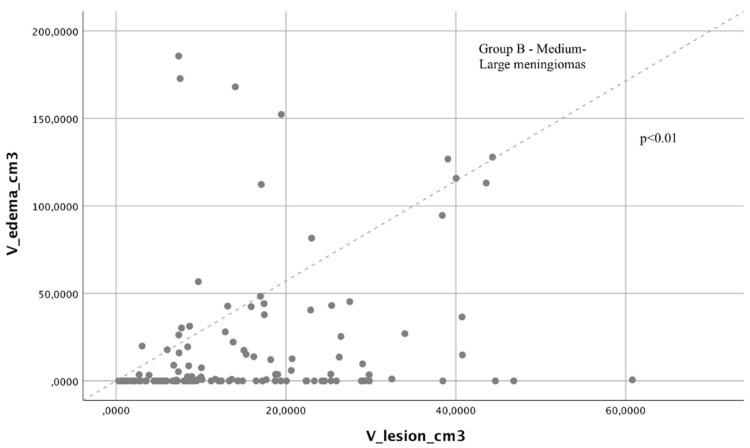
Through an ANOVA study, we found statistically significant differences between the reduction in performance in the postoperative phase and at the last evaluation among the various locations of GIMs.

**Figure 5 brainsci-12-00817-f005:**
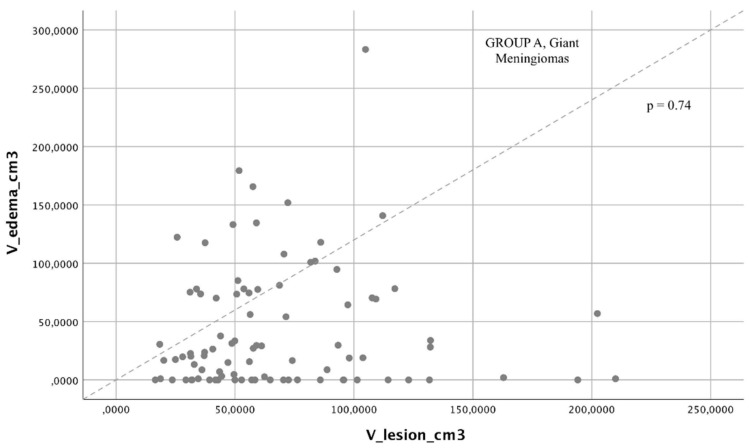
Chi-square comparison analysis between the two groups analyzed: giant meningiomas had a higher rate of postoperative complications than a comparable mortality rate compared with medium/large meningiomas.

**Figure 6 brainsci-12-00817-f006:**
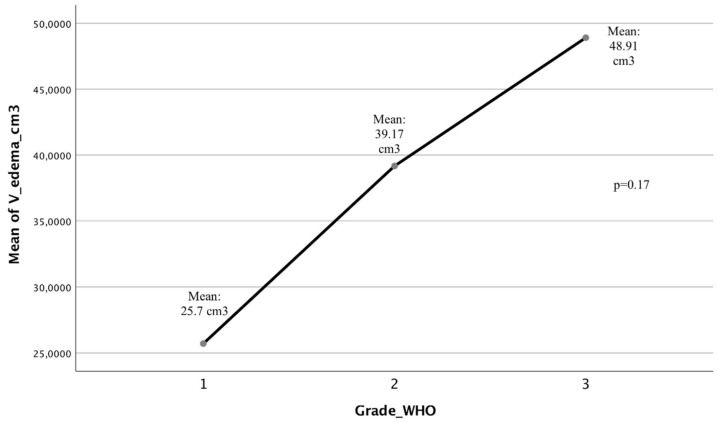
The one-way ANOVA study demonstrates a linear relationship between edema volume and meningioma grading, with low significance (*p* = 0.167).

**Figure 7 brainsci-12-00817-f007:**
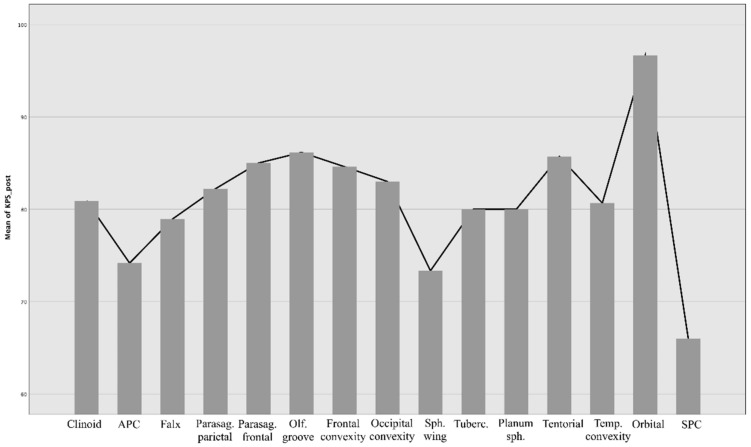
Measures of outcome, reported as KPS value, in the early postoperative phase according to different locations.

**Figure 8 brainsci-12-00817-f008:**
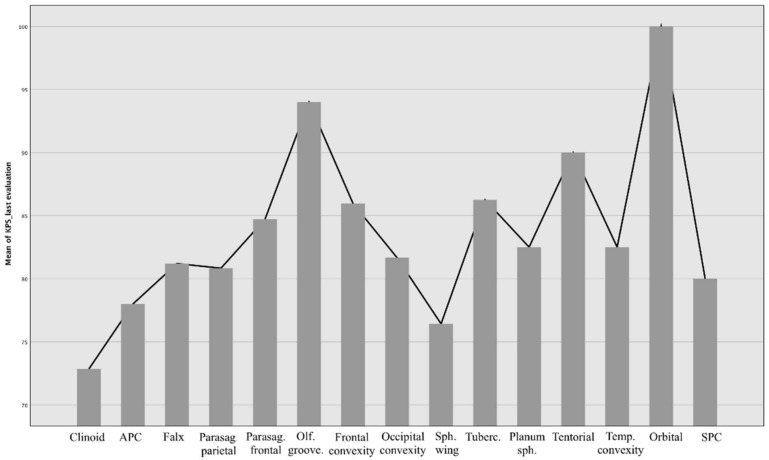
The same measures of outcome (using KPS value), reported at last evaluation (after 12 months at least).

**Figure 9 brainsci-12-00817-f009:**
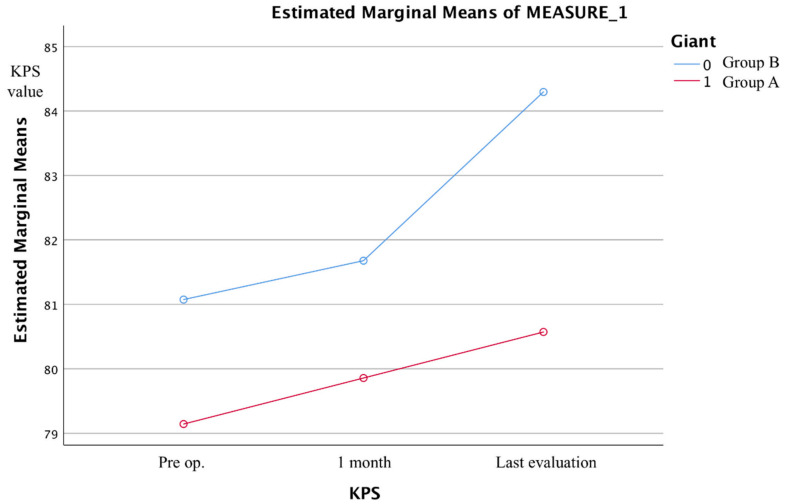
Results of a repeated measures test on how GIMs, independently from location and clinical debut, maintain a stable reduced performance status after surgery.

**Figure 10 brainsci-12-00817-f010:**
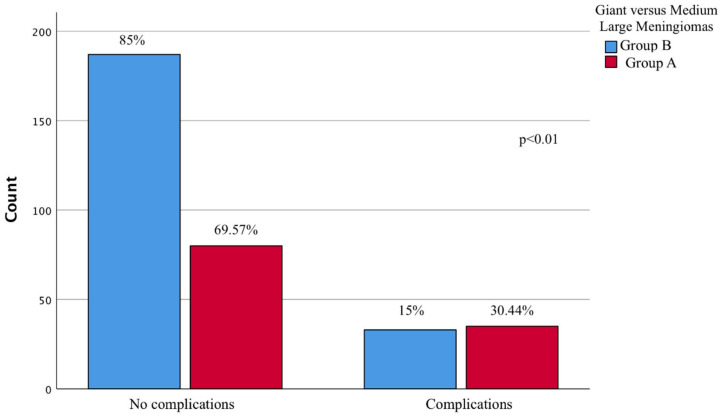
This bar graph shows that GIMs showed a higher surgical risk of complication than medium/large meningiomas.

**Figure 11 brainsci-12-00817-f011:**
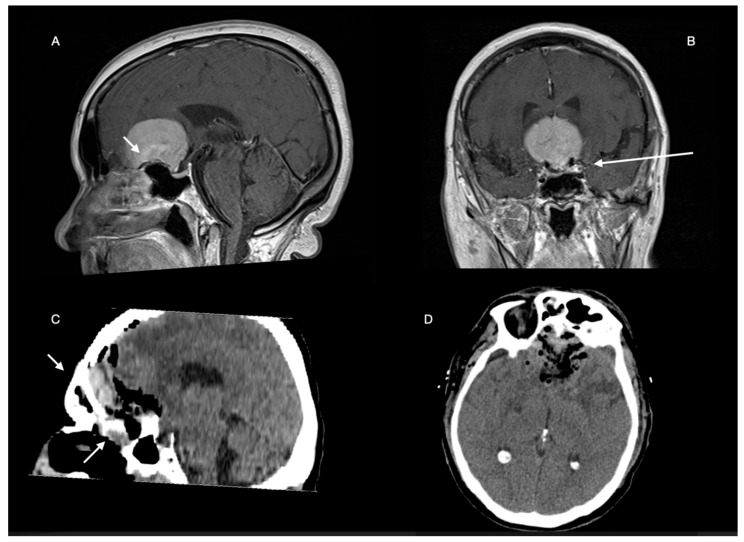
We present a case of a 66-year-old woman with frontobasal meningioma. On MRI (**A**,**B**), there is partial bilateral encasement of the anterior cerebral arteries and partial erosion of the floor of the anterior cranial fossa (white arrows). Surgical excision resulted in partial inferior residual and compromised anterior fossa floor (white arrows, **C**,**D**). The patient after an initial postoperative recovery showed rhinoliquorrhea after about 4 days, which was subsequently treated endoscopically. The patient underwent prolonged antibiotic treatment, leading to a delay in subsequent therapy (radio surgery on residual).

**Table 1 brainsci-12-00817-t001:** The population study.

Final Series	No. 340	
**Age**	Mean: 60.38 Median: 62 Sd: 13.56	Min: 20 Max: 90
**Sex (Female)**	F: 238–70%	
**Smoke**	98 = 28.8%	
**Hypertension**	108 = 31.8%	
**Clinical debut**	Incidental (1) = 45–13.2%	Headache (4) = 46–13.5%
	Dizziness (2) = 32–9.4%	Seizure (5) = 88–25.9%
	Focal deficit (3) = 80–23.5%	Mental alteration (6) = 46–13.5%
**Hospitalization (330 pts)**	Mean: 17.76 Median: 13 Sd: 17.23	Min: 5 Max: 209
**Follow-up (months)**	Mean: 47.76 Median: 47 Sd: 14.82	Min: 12 Max: 72
**WHO grade**	Grade I: 285–83.8%	
	Grade II: 47–13.8%	
	Grade III: 8–2.4%	
**Histological type**	1 = Meningothelial–205–60.3%	7 = Secretory–12–3.5%
	2 = Psammomatose–16–4.7%	8 = Anaplastic–7–2.1%
	3 = Transitional–22–6.5%	9 = Angiomatous–9–2.6%
	4 = Microcystic–8–2.4%	10 = Lymphoplasmacitic–1–0.3%
	5 = Atypic–40–11.8%	11 = Metaplastic–5–1.5%
	6 = Fibrous–13–3.8%	
**Switch in a malignant form**	5/335–1.5%	
**Multiple/meningiomatosis**	13 pts–3.8%	
**Location/position**	1 = Clinoid–11–3.2%	9 = Sphenoid wing–20–5.9%
	2 = APC–12–3.5%	10 = Tuberculum sellae–9–2.6%
	3 = Falx–39–11.5%	11 = Planum sphenoidal–8–2.4%
	4 = Parasagittal parietal–21–6.2%	12 = Tentorial–15–4.4%
	5 = Parasagittal frontal–26–7.6%	13 = Temporal convexity–15–4.4%
	6 = Olfactory groove–14–4.1%	14 = Orbital–3–0.9%
	7 = Frontal convexity–85–25%	15 = Sphenopetroclival–12–3.5%
		Subtentorial–26–7.6%

**Table 2 brainsci-12-00817-t002:** The main clinical, radiological, and outcome variables examined in the study comparing giant and medium/large meningiomas.

Groups	Giant Meningiomas: 117 pts	Medium/Large Meningiomas: 223 pts	*p*-Value
**Sex**	M: 46–39.3%	M: 56–25.1%	1.00
	F: 71–60.7%	F: 167–74.9%	
**Age**	Min: 20 Max: 90	Min: 25 Max: 89	1.00
	Mean: 60.62	Mean: 60.26	
	Median: 64	Median: 60.50	
	SD: 13.99	SD: 13.35	
**Smoke**	37 = 31.6%	61 = 27.4%	0.24
**Hypertension**	42 = 35.9%	66 = 29.6%	0.24
**Comorbidity CCI scale (points)**	0 = 25	0 = 44	
	1 = 21	1 = 36	1.00
	2 = 24	2 = 47	1.00
	3 = 16	3 = 39	0.23
	4 = 21	4 = 31	0.98
	>5 = 10	>5 = 26	0.67
**Seizure at onset**	24 pts = 20.5%	49 pts = 22%	0.76
**WHO grade**	Grade I: 81–69.2%	Grade I: 204–91.5%	1.00
	Grade II: 31–26.5%	Grade II: 16–7.2%.	<0.01
	Grade III: 5–4.3%	Grade III: 3–1.3%	0.09
**Maximum diameter (cm)**	Min: 5 Max: 10.5	Min: 0.80 Max: 4.9	
	Mean: 6.26	Mean: 3.3	
	Median: 6	Median: 3.3	
**V edema cm^3^**	Mean: 42.52 SD: 52.77	Mean: 18.37 SD: 38.59	<0.01
**V lesion cm^3^**	Mean: 67.32 SD: 39.31	Mean: 15.79 SD: 12.59	<0.01
**Mitotic index/10 HPF**	Mean: 1.92 SD: 2.4	Mean: 1.2 SD: 1.9	<0.01
**Ki-67 expression**	Mean = 7%	Mean = 4.5%	0.02
**PR+**	12 pts	26 pts	0.42
**Simpson grade resection**	1 = 51 pts–43.6%	1 = 102 pts–45.6%	
	2 = 20 pts–17.1%	2 = 35 pts–15.7%	
	3 = 5 pts–4.3%	3 = 7 pts–3.1%	
	4 = 1 pts–0.9%	4 = 1 pts–0.7%	
**Hospitalization**	Mean = 18.71	Mean = 17.27	0.48
**Complications**	35 pts: 29.9%	33 pts: 14.8%	<0.01
**Complications**	Hydrocephalus = 2 pts–1.7%	Hydrocephalus = 5 pts–2.2%	1.00
	Hemorrhage= 2 pts–1.7%	Hemorrhage= 4 pts–1.8%	1.00
	Infections= 16 pts–13.7%	Infections= 10–4.5%	<0.01
	Intractable seizure = 5 pts–4.3%	Intractable seizure = 6–2.7%	1.00
	Ischemia = 10 pts–8.5%	Ischemia = 8–3.6%	0.05
**Recurrence**	17 pts = 14.5%	20 pts = 9%	0.11
**Dead at last evaluation**	14 pts: 12%	16 pts: 7.2%	0.14
**KPS at onset**	Mean = 70–80 DS= 14.72	Mean = 80 DS = 14	0.57
**KPS after procedure**	Mean = 80 DS = 20	Mean = 80 DS = 10	1.00
**KPS last evaluation**	Mean = 80	Mean = 80–90	0.12

## Data Availability

The dataset generated and analyses during the current study are not publicly available and are not available in national databases, because they are the result of a institutional internal research of all treated cases of IM in our hospital (Policlinico Umberto I of Rome and Spaziani Hospital of Frosinone). The original dataset is available from the Corresponding Author on reasonable request.

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
