# Peer review of "The Surgical Risk Factors of Giant Intracranial Meningiomas: A Multi-Centric Retrospective Analysis of Large Case Serie"

_brainsci, 2022, doi:10.3390/brainsci12070817_

Round 1

Reviewer 1 Report

The authors present a significant case history of patients undergoing intracranial meningioma excision surgery. They compared two groups (giant and medium/large) with a careful and interesting statistical analysis. The authors should provide a PRISMA to better define the pathway for inclusion of cases. I suggest clarifying the definition of medium and large meningioma and further differentiating patients in the medium/large group into 2 groups. 

Page 2: "Patients with small Meningiomas (less than 2cm) was submitted to a surgery if symp tomatic or if they demonstrated failure in radio surgery or medical therapy for seizure." clarify that these patients were not included in the analysis. Figure 3 refers to 3 cm lesions specify the meaning of this cut off. authors should provide more pictures and case descriptions. describe the characteristics of patients with complications and protracted hospitalization. finally, I suggest categorizing patients with comorbidity scale (Charlson for example)

Author Response

The authors present a significant case history of patients undergoing intracranial meningioma excision surgery. They compared two groups (giant and medium/large) with a careful and interesting statistical analysis. 

R: We want to thank you for the consideration of our manuscript, and for the great opportunity to improve our work. We present here a point-to-point response.

The authors should provide a PRISMA to better define the pathway for inclusion of cases. I suggest clarifying the definition of medium and large meningioma and further differentiating patients in the medium/large group into 2 groups. 

R: PRISMA is a useful tool for patient selection in META-ANALYSIS and SYSTEMATIC REVIEW, and this is a retrospective clinical study. Assuming, however, that the reviewer intended a flow-chart useful for understanding the final selection of the patients to be analyzed, we show among the images the diagram with the patients excluded at various stages. In defining the size of meningiomas, we referred to the traditional classification of Russell et. al. [60] which reports small meningiomas as those less than 2 cm in diameter, medium as 2 to 4 cm, and large up to 4.9 cm. Patients with small Meningiomas (less than 2cm and normally submitted just to follow-up or radio surgery) was submitted to a surgery if symptomatic or if they demonstrated failure in radio surgery or medical therapy for seizure. Medium/large meningiomas were considered as one group for this study because they have similar prognostic features and are often variably defined in many other studies. This sentence was added in the manuscript

Page 2: "Patients with small Meningiomas (less than 2cm) was submitted to a surgery if symp tomatic or if they demonstrated failure in radio surgery or medical therapy for seizure." clarify that these patients were not included in the analysis. 

R: The patients included were only those who actually had surgery with the aim of removing the tumor. The sentence only explains why there are 2-cm tumors included in the case series that are normally sent for follow-up or radio surgery only. We clarified this point better in the text.

Figure 3 refers to 3 cm lesions specify the meaning of this cut off. 

R: We are sorry but it is not possible to correct this wording. The diagram is taken directly from SPSS and cm3 means cm3 . It is not possible to add the superscript on the software.

Authors should provide more pictures and case descriptions. describe the characteristics of patients with complications and protracted hospitalization. 

R: We provided to add in the text a representative case of frontal located GIM complicated after surgery.

I suggest categorizing patients with comorbidity scale (Charlson for example)

R: We thank you very much for this suggestion. We were thus able to consider much more anamnestic data and make the analysis more complete. We found no significant differences between the two groups for the higher Charlson scores, and we have included the analysis in the text and Table 2.

Reviewer 2 Report

1. The manuscript was not prepared according to the journal's requirements, for example in the introduction the second ref is no 49 etc. 

2. there is no description for figure 2 under the figure. 

3. in the figure 1 please indicate the described structures.

5. I cannot see the keywords

6. all sections were described not good.

7. the major criticism is associated to lack of data of bioethical committee in this study.

8. tables and figures are not self-explanatory.

9. however, the title "says" about risk factors, the authors did not present any OR.

10. discussion is boring.

Author Response

We would like to thank the reviewer, for his helpful suggestions and help in improving our work.

  1. The manuscript was not prepared according to the journal's requirements, for example in the introduction the second ref is no 49 etc. 

R: We addressed the references in numerical order.

2. there is no description for figure 2 under the figure. 

R: We disposed the figure caption and description under every picture.

3. in the figure 1 please indicate the described structures.

R: we added white arrows to indicate structures and analysis and described it in the captions

5. I cannot see the keywords

R: they are on top at first page

All sections were described not good, the major criticism is associated to lack of data of bioethical committee in this study.

R: This study represents the retrospective arm of a larger study involving the evaluation of surgical cases and La Sapienza University. The institutional review board IRB 6168 Prot. 0935/2020

8. tables and figures are not self-explanatory.

R: we improved descriptions

10. discussion is boring.

R:Unfortunately, with peer review defined as "boring," we do not have sufficient evidence to conduct an accurate review of the discussion. However, we have tried to streamline it and make it shorter and focused on the results.

Reviewer 3 Report

The multi-centric retrospective analysis by Armocida et al. aims to investigate whether the presence of large diameter and peritumoral brain edema (PBE) on radiological diagnosis indicates different mortality rates, grading, characteristics, and outcomes. The authors found Giant intracranial meningiomas (GIMs) have a higher risk of developing complications in the postoperative phase than medium/large meningioma without higher risk of mortality and recurrence. The study is interesting though some critical issues would need to be addressed.

General concept comments

1. The authors stated that GIMs have a higher risk of developing complications in the postoperative phase, which seems reasonable since the surgery is more complex, giving rise to severer injury to the patients. I am a bit doubting the significance of this study.

2.  In addition to the tumor size, it is very important to know whether the location is also associated with the postoperative complications, which is not taken into account in the manuscript.

Specific comments 

1. The manuscript is not well written and needs proofreading by a native English speaker.

2. The data presented in the tables are not in a well-structured manner.

3. The keywords are missing.

Author Response

The multi-centric retrospective analysis by Armocida et al. aims to investigate whether the presence of large diameter and peritumoral brain edema (PBE) on radiological diagnosis indicates different mortality rates, grading, characteristics, and outcomes. The authors found Giant intracranial meningiomas (GIMs) have a higher risk of developing complications in the postoperative phase than medium/large meningioma without higher risk of mortality and recurrence. The study is interesting though some critical issues would need to be addressed.

R: We thank the editor and reviewers for the insightful, valuable, and thoughtful comments, which gave us the opportunity to improve our paper. Some suggestions are unfortunately irreconcilable and with conflicting opinions. For example, according to some reviewers, the methods section is too long, while others require it to be implemented by describing the surgical steps for each approach or add further statistical analysis.  Therefore, we preferred to maintain the completeness in order to ensure reproducibility in the study in case you want to set up a prospective study rather than excessively cut the methodological part (reduced and trimmed in any case of redundant parts).

General concept comments

  1. The authors stated that GIMs have a higher risk of developing complications in the postoperative phase, which seems reasonable since the surgery is more complex, giving rise to severer injury to the patients. I am a bit doubting the significance of this study.

R: we added more graphs and analysis to show our results.

2.  In addition to the tumor size, it is very important to know whether the location is also associated with the postoperative complications, which is not taken into account in the manuscript.

R: Our study showed that indeed there are some locations more frequently prone to complications than others. Among them, we obtained relevant results regarding meningiomas of the anterior basicranium. We preferred given the vastness of the data and analysis on so many patients to focus on the significant differences there are between giant and medium/large meningiomas by analyzing mainly postoperative functional recovery. The data that the reviewer rightly noted were useful for completeness will be published soon in another paper.

Specific comments 

  1. The manuscript is not well written and needs proofreading by a native English speaker.

R: The manuscript has now undergone editing using professional software for English

2. The data presented in the tables are not in a well-structured manner.

R_ We edited the tables

3. The keywords are missing.

R: Keywords added

Round 2

Reviewer 1 Report

The authors revised the work as suggested

Reviewer 2 Report

1. Unfortunately, the manuscript has not been corrected by the authors as recommended. Even the article template was not used, and the necessary formatting was not carried out.

2. The captions under the figures are incorrect, eg under figure 10 the text looks more like a description of the chart, not its title; in the case of figure 11, the caption is placed above and below the figure.

3. The work was prepared and "corrected" in a very sloppy manner.

Reviewer 3 Report

The authors have adequately addressed all of my concerns.